# Socioeconomic Status, Parental Education, School Connectedness and Individual Socio-Cultural Resources in Vulnerability for Drug Use among Students

**DOI:** 10.3390/ijerph17041306

**Published:** 2020-02-18

**Authors:** Gilberto Gerra, Elisa Benedetti, Giuliano Resce, Roberta Potente, Arianna Cutilli, Sabrina Molinaro

**Affiliations:** 1Drug Prevention and Health Branch, Division for Operations, United Nations Office on Drugs and Crime, P.O. Box 500, 1400 Vienna, Austria; gilberto.gerra@un.org; 2Epidemiology and Health Research Lab, Institute of Clinical Physiology—IFC, National Research Council of Italy—CNR, Via G. Moruzzi, 1, 56124 Pisa, Italy; elisa.benedetti@ifc.cnr.it (E.B.); giuliano.resce@ifc.cnr.it (G.R.); roberta.potente@ifc.cnr.it (R.P.); arianna.cutilli@ifc.cnr.it (A.C.)

**Keywords:** socioeconomic inequalities, parental education, drug use, cannabis, cocaine, heroine

## Abstract

Background and Aims: Families who live in a disadvantaged socioeconomic situation frequently face substandard housing, unsafe neighborhoods, inadequate schools and more stress in their daily lives than more affluent families, with a host of psychological and developmental consequences that can hinder their children’s development in many ways. However, the measurement of socioeconomic status among youth and its link with different forms of illicit substance use is challenging and still unclear. This paper extends existing research on the relationship between socioeconomic status and illicit drug use among adolescents by focusing on three different patterns of use (experimental, episodic and frequent) and making use of two indicators to improve the measurement of individual socioeconomic characteristics in a big sample of European students. Methods: Data were drawn from the European school Survey Project on Alcohol and other Drugs (ESPAD), which, since 1995, collects comparable data among 15-to-16-year-old students to monitor trends in drug use and other risk behaviors across Europe. The sample comes from 28 countries that participated in the 2015 data collection. The consumption of cannabis, cocaine and heroin are considered, and the related patterns are identified based on the frequency of use. Family characteristics at student level are defined through two dimensions: parental educational level and perceived socioeconomic status. Multivariate multilevel mixed-effects logistic regression was performed in order to measure the association between individual characteristics and vulnerability for drug use. Results: Some patterns of use, episodic and frequent in particular, were found strongly associated with a lower socioeconomic status and lower parental education. Conclusions: Our results suggest that drug policies should be combined with actions aimed at removing barriers to social inclusion that are attributable to the socioeconomic background of adolescents.

## 1. Introduction

Childhood socioeconomic status and school failure have been found to predict drug use in youth and young adults [1], suggesting an association between childhood social disadvantages and later use of psychoactive drugs, primarily cannabis [2,3,4,5]. In line with these findings, recent evidence has indicated that minorities with lower socioeconomic status had higher prevalence of lifetime use of marijuana, and higher incidence of past year initiation compared with affluent social groups in the population [6,7]. However, the link between socioeconomic disadvantage and drug use is very complex. A French study found that adolescents from affluent families are more prone to experimentation with cannabis, but the heavy use is associated with lower family socioeconomic status and school failure [8].

The impact of family affluence on drug use was recently exhibited within a dynamic interaction with other factors. The expected drop in frequency of drug use among youth from rich families was observed among young people with expectations of academic achievements. In contrast, among affluent youth with lower academic expectations and poor school performance, the high socioeconomic status seems to become an additional risk factor [9].

The drug use seems mediated by a variety of adversities, particularly those related to education inequalities [10,11,12,13]. Compared with their higher-income counterparts, children growing up in low-income families were reported to complete less schooling and achieve lower results, report worse health, and both work and earn less in adulthood [13,14]. Additionally, children living in poverty tend to be concentrated in low-performing schools staffed by ill-equipped teachers, in turn aggravating social disadvantages [15,16].

This paper contributes to extend existing research on the relationships between socioeconomic status and illicit drug use among adolescents, by distinguishing different patterns of use (experimental, episodic and frequent use) and making use of two different indicators to more accurately measure the individual socioeconomic status in a big sample of European adolescents (European school Survey Project on Alcohol and other Drugs—ESPAD).

Considering previous works which analyze the interaction between social factors and psychoactive substance abuse in ESPAD, Perelman et al. [17] showed an association between heavy smoking and school absenteeism among youth. Shackleton et al. [18] proved that, although there are large country-level differences in socioeconomic inequalities and adolescent substance use, young people with lower socioeconomic status have a significantly higher odds of heavy episodic drinking, regular smoking and cannabis use. Considering only the Danish students, König et al. [19] showed that higher school performance is related to lower alcohol consumption, but low sociodemographic status is not associated with higher alcohol consumption. Using the Finnish surveys from 1999 to 2015, Raitasalo et al. [20] evidence that the decline in alcohol use and heavy episodic drinking among youth is associated with the reduced availability of alcohol, an increase in parental monitoring, and with the introduction of new digital technologies and new forms of interaction within families and peer groups. Compared with this previous evidence drawn from ESPAD, the distinction of three different patterns of use (experimental, episodic and frequent) of three different illicit drugs (cannabis, cocaine and heroin) in the present study allows us to analyze the different association that social factors may have with different students’ patterns of use. As the students’ experimental, episodic, and frequent use of cannabis, cocaine and heroin are heterogeneous risk behaviors, this study allows us to capture a differentiated association with socioeconomic status of the family, parental education, school connectedness and individual socio-cultural resources.

The rest of paper is as follows: Section 2 presents the methods, Section 3 contains results, Section 4 discusses the results and Section 5 draws the conclusions. 

## 2. Materials and Methods 

### 2.1. Data and Design

Data for the present study were drawn from the ESPAD cross-sectional survey, which, since 1995, collects comparable data among 16-year-old students to monitor trends in drug use and other risk behaviors within and between European countries. The sample (Male = 24,136; Female = 26,300) comes from 28 out of the 35 countries that participated in the 2015 data collection: Albania, Austria, Belgium (Flanders), Bulgaria, Croatia, Cyprus, Czech Republic, Faroe Islands, France, Macedonia (FYR of), Georgia, Germany (Bavaria), Greece, Hungary, Iceland, Ireland, Italy, Latvia, Liechtenstein, Lithuania, Malta, Moldova, Montenegro, Netherlands, Poland, Romania, Slovenia and Ukraine. The methodology used national samples of randomly selected schools/classes in which the cohort of students born in 1999 completed the standardized ESPAD questionnaire. Participating countries adhered to common research guidelines to guarantee consistency in sampling, questionnaires and survey implementation, and confirmed to the respective national ethics and data protection regulation. All samples are nationally representative, apart from Belgium (only the Flanders region), Cyprus (only government-controlled areas) and Moldova (the Transnistria region is not included). Details about sampling, data collection methodology and ethics in each country are reported in Kraus et al. [21] and Guttormsson et al. [22]. An overview of the geographical coverage, sampling procedure in each country, representativeness of the samples and characteristics of the samples are provided in Tables C and F–H of the ESPAD methodology Report [22], pp. 10, 16, 18, 29–30.

### 2.2. Dependent Variables

#### 2.2.1. Cannabis

In the case of cannabis consumption, in order to identify which patterns of use are more affected by the student’s characteristics, we identified three categories of users: Experimenters: (1) those students having tried the drug only once or twice in their lifetime and (0) otherwise;Episodic users: (1) those students who used the drug more than twice in their lifetime, but less than 20 times in in the past month and (0) otherwise;Frequent users: (1) those students having used the drug at least 20 times in past month and (0) otherwise.

#### 2.2.2. Cocaine and Heroin

Cocaine and heroin were analyzed separately, but we used the same categorization for the frequency, as follows: Experimenters: (1) those students having tried at least once, but not more than twice in their lifetime and (0) otherwise;Episodic users: (1) those students who used more than twice, but less than 20 times in their lifetime and (0) otherwise;Frequent users: (1) those students reporting having used 20 times or more and (0) otherwise.

### 2.3. Independent Variables

#### 2.3.1. Parental Education

Parental education was assessed by considering the highest level of education of the student’s parents (as in the Economic Social and Cultural Status Index in the Programme for International Student Assessment—PISA project of the Organisation for Economic Co-operation and Development—OECD, see OECD, 2016). In the ESPAD questionnaire, students are asked, “What is the highest level of schooling your father completed?” and “What is the highest level of schooling your mother completed?” separately. We first select the highest level of education between father and mother, and then we dichotomized the five options by which the students can answer, as follows: completed primary school or less/some secondary school (1) and completed secondary school/some college or university/completed college or university (0).

#### 2.3.2. Socioeconomic Status of the Family

The socioeconomic status of the family is investigated in ESPAD questionnaire by means of the following question: “How well-off is your family compared to other families in your country?”. We dichotomized the seven options by which the students can answer as follows: very much better off/much better off/Better off/About the same (0) and less well-off/much less well-off/very much less well off (1).

#### 2.3.3. Truancy at School 

Truancy at school was included as a proxy for school connectedness. School connectedness should be taken into account since previous studies have shown that the adoption of risk behaviors among youth is significantly linked to school absenteeism [17,19]. We consider the number of days in the past month in which students have missed one or more lessons because they skipped or ‘cut’. In the ESPAD questionnaire there are six answer options for this question: none; 1 day; 2 days; 3–4 days; 5–6 days; and 7 days +. We dichotomize this outcome as follows: none/1 day/2 days (0) and 3–4 days/5–6 days/7 days + (1).

#### 2.3.4. Reading books for enjoyment 

We also included a proxy for individual socio-cultural resources by using the fact of reading books for pleasure as a signal of engagement in cultural leisure activities. It has been shown that socioeconomic status heavily influences access to relevant networks (e.g., internet, newspapers and libraries) for socio-cultural resources [23], which in turn seems to become an additional risk factor for substance use [9,12]. Students are asked how often they read books for enjoyment (schoolbooks are excluded). The options by which they can answer were classified as follows: at least one book per month (0), and less than one book per month (1).

### 2.4. Statistical Analysis

Multivariate multilevel mixed-effects logistic regression was performed in order to measure the association between perceived socioeconomic status of the family, parental education and the difference frequencies of cannabis, cocaine and heroin. All analyses are adjusted for gender and country-level Gross Domestic Product (GDP). The first is included to control for possible gender differences, as the existence of a gender gap has been shown in many substance-use behaviors, with females showing lower rates of use (see ESPAD Report 2015 as a reference). Country-level GDP has been included, to control for country-level differences in GDP levels that may confound the effect of individual level differences in the perceived individual SES.

Models were performed on the overall sample, modeling different countries as random effects as in Molinaro et al. [24]. The multilevel model allows the inclusion of both levels (student and country) in the same analysis, avoiding bias due to correlation between students within the same country. The data have a hierarchical structure where students’ characteristics (level 1) are nested in the country (level 2), with the likelihood that students’ pattern of use of cannabis, cocaine, and heroine is correlated with belonging to the country where they live. As discussed recently by Stevens [25], when there is heterogeneity in the relationship between an independent variable and a dependent variable between the units at level 2 of the model, then it is usual to include both the random and the fixed effect of that variable in order to improve goodness of fit of the model to the data. We investigate the determinants of probable substance use by means of adjusted odds ratios (aORs) with a 95% confidence interval. All the statistical analyses were performed by using R [26]. 

In order to investigate the possible effect of school connectedness and individual socio-cultural resources, we analyzed the relationship between socioeconomic status and parental education with drug use by two models: -Model A includes students’ socioeconomic status of the family and parental education, but it does not include truancy at school and reading books for enjoinment;-Model B includes students’ socioeconomic status of the family and parental education, and it also considers truancy at school and reading books for enjoinment.

## 3. Results

As shown in Table 1, the average size of sample is 1801 students for Country; among them, regarding cannabis, the prevalence of episodic users is 8.66%, experimenters are 6.54% and frequent users 0.72% in average. Regarding cocaine use, the prevalence of episodic use is 0.53%, experimenters are 1.13% and frequent users are 0.22% in average. As far as heroin is concerned, the prevalence of episodic users is 0.34%, experimenters are 0.46% and frequent users are 0.11% in average.

### 3.1. Cannabis

As shown in Figure 1A, low socioeconomic status of the family is significantly associated with the episodic (aOR = 1.38, 95% CI = 1.24–1.54) and the frequent use (aOR = 2.38, 95% CI = 1.79–3.16) of cannabis, but it is not significantly associated with the experimental use of cannabis. Low parental education is not significantly associated with either the experimental, the episodic or the frequent use of cannabis. 

Figure 1B shows that truancy at school is significantly associated with the experimental use (aOR = 1.71, 95% CI = 1.49–1.96), the episodic use (aOR = 3.81, 95% CI = 3.42–4.25) and the frequent use (aOR = 7.31, 95% CI = 5.53–9.67) of cannabis. Additionally, not reading books for enjoyment is also significantly associated with the experimental use (aOR = 1.19, 95% CI = 1.09–1.29), the episodic use (aOR = 1.27, 95% CI = 1.18–1.37) and the frequent use (aOR = 2.24, 95% CI = 1.70–2.95) of cannabis. 

Figure 1B shows that, even after controlling for truancy at school and reading books for enjoyment, low socioeconomic status of the family maintains a significant association with the episodic use (aOR = 1.32, 95% CI = 1.18–1.48) and the frequent use (aOR = 2.14, 95% CI = 1.60–2.85) of cannabis, but it is not significantly associated with the experimental use of cannabis. The inclusion of truancy at school and reading books for enjoyment has reduced the aOR for both episodic use and frequent use of cannabis (from 1.38 to 1.32 and from 2.38 to 2.14, respectively). 

### 3.2. Cocaine

Figure 2A shows that low socioeconomic status of the family is significantly associated with the experimental use (aOR = 1.48, 95% CI = 1.14–1.91) and the episodic use (aOR = 2.09, 95% CI = 1.49–2.92) of cocaine, but it is not significantly associated with the frequent use. Low parental education is significantly associated with the experimental use (aOR = 1.37, 95% CI = 1.07–1.76) and the frequent use (aOR = 2.28, 95% CI = 1.35–3.84) of cocaine, but it is not significantly associated with the episodic use. 

Figure 2B shows that truancy at school is significantly associated with the experimental use (aOR = 5.16, 95% CI = 4.20–6.34), the episodic use (aOR = 6.29, 95% CI = 4.69–8.44) and the frequent use (aOR = 6.75, 95% CI = 4.35–10.49) of cocaine. Not reading books for enjoyment is significantly associated with the episodic use (aOR = 1.34, 95% CI = 1.01–1.77) and the frequent use (aOR = 2.23, 95% CI = 1.38–3.61), but not with the experimental use, of cocaine. 

Figure 2B also shows that, even after controlling for truancy at school and not reading books for enjoyment, low socioeconomic status of the family maintains a significant association with the experimental use (aOR = 1.38, 95% CI = 1.06–1.78) and the episodic use (aOR = 1.94, 95% CI = 1.39–2.73) of cocaine, but it is not significantly associated with the frequent use. The inclusion of truancy at school and not reading books for enjoyment has reduced the aOR for both the experimental and the episodic use of cocaine (from 1.48 to 1.38 and from 2.09 to 1.94, respectively). After controlling for truancy at school and reading books for enjoyment, low parental education remains significantly associated with the experimental use (aOR = 1.28, 95% CI = 1.01–1.63) and the frequent use (aOR = 1.91, 95% CI = 1.14–3.21) of cocaine, but it is not significantly associated with the episodic use. The inclusion of truancy at school and reading books for enjoyment has reduced the intensity of the associations between parental education and the pattern of use of cocaine (aOR from 1.37 to 1.28 for the experimental use and aOR from 2.28 to 1.91 for the frequent use).

### 3.3. Heroin

Figure 3A shows that low socioeconomic status of the family is not significantly associated with either the experimental, the episodic or the frequent use of heroin. Nevertheless, low parental education is significantly associated with the experimental use (aOR = 1.52, 95% CI = 1.03–2.23) and the frequent use (aOR = 2.50, 95% CI = 1.20–5.24) of heroin, but it is not significantly associated with the episodic use. Figure 3B provides evidence that truancy at school is significantly associated with the experimental use (aOR = 4.78, 95% CI = 3.49–6.54), the episodic use (aOR = 6.55, 95% CI = 4.65–9.22) and the frequent use (aOR = 8.61, 95% CI = 4.67–15.86) of heroin. Not reading books for enjoyment is not significantly associated with either the experimental use, episodic use or the frequent use. 

After controlling for truancy at school and reading books for enjoyment, low parental education remains significantly associated with the frequent use of heroin (aOR = 2.15, 95% CI = 1.03–4.51), although with a lower intensity (from 2.5 to 2.15). No significant association is found between low parental education and the experimental and the episodic use. 

### 3.4. A global Perspective

In Figure 4, all the aORs described in Figure 1A, Figure 2A and Figure 3A are shown together. On the axis the aOR defining the association between low socioeconomic status of the family and the different drug-use patterns is reported, while on the axis, the aOR defining the association between low parental education and the different drug use patterns is reported. Moreover, aORs shown in Figure 4 are adjusted for gender and country-level Gross Domestic Product (GDP). Drawing two lines along the value 1 (no associations) allows us to get four different quadrants: the upper right quadrant with low socioeconomic status of the family and low parental education; the bottom right quadrant with high socioeconomic status of the family and low parental education; the bottom left quadrant with high socioeconomic status of the family and high parental education; and the upper left quadrant with low socioeconomic status of the family and high parental education. 

The majority of drug-use patterns are concentrated in the upper right quadrant. On the contrary, none of them is located either in the bottom left quadrant or in the bottom right quadrant. Interestingly, all the patterns of use of cannabis are located in the upper left quadrant. 

In Figure 5, the same analysis shown in Figure 4 is repeated, using the aORs described in Figure 1B, Figure 2B and Figure 3B (model with the inclusion of “truancy at school” and “reading books”, in addition to gender and per-capita GDP among independent variables). Overall, the same evidence shown in Figure 4 is confirmed in Figure 5, but one exception is worthy of attention: The frequent use of cocaine is associated with high socioeconomic status with model B. Although not significant (see Figure 2B), this may be attributed to the high price of cocaine, making it not affordable for youth from family with low socioeconomic status [27]. 

## 4. Discussion

In line with previous studies on the association between social factors and drug abuse among students (for the ESPAD case see Perelman et al. [17], Shackleton et al. [18], König et al. [19], and Raitasalo et al. [20]), our results suggest that some patterns of use, episodic and frequent in particular, are associated with a lower socioeconomic status and lower parental education. Specifically, our findings are in line with the findings of previous studies indicating that the adolescents from affluent families were at high risk for cannabis experimentation, but appeared to be less prone to engage in daily use [5,8,28,29,30].

Low SES in our study was found to be associated with experimenting and episodic use of cocaine, but not with frequent use. This may be attributed to the high price of cocaine, making it not affordable for youth [30].

In our study, low parental education attainment was not significantly associated with cannabis experimenting, episodic or frequent use. In contrast, low level of education of the parents was significantly associated with the frequent use of cocaine and heroin. 

Considering the favorable public opinion and perception of cannabis acceptability, combined with higher prevalence of cannabis consumption with respect to cocaine and heroin, it could be argued that low level parental education would be a specific risk co-factor only in the families with severe vulnerability conditions and in a framework of poor socio-cultural resources. High frequency of cocaine and heroin use characterized the adolescents with parents with low education attainment, as recently reported in the literature [31,32,33].

A strongly significant association between truancy, expressed as number of missed school days (no motivation to go), and the probability of frequent cannabis, cocaine and heroin use among our adolescents was demonstrated in our sample. Truancy was considered to be a measure of low school connectedness, already reported in previous studies in association with drug-use vulnerability. Particularly, prospective evidence supporting the protective effects of school connectedness with respect to drug use has been previously obtained by other research groups [34]. The adolescents living and growing in low-resource settings who disliked school were already known to be at a greater risk of adult drug use [35]. Focusing on ESPAD, a significant association has been shown between school absenteeism and smoking intensity among youth [17]. In general, connectedness to school during adolescence has been shown by previous studies to be a protective factor for lower rates of health-risk behaviors, comprising substance use [36,37,38,39,40,41,42,43]. The present study confirms this evidence. Furthermore, concerning control variables, the protective role of being female [21], in relation to substance use, is confirmed by the present analysis, whilst country-level GDP does not seem to play a role. 

Overall, the results of our analyses feed into the more general framework of the study of inequality of opportunities [44]. 

This work has some limitations that need to be discussed. First, the study is based on a single data source, i.e., ESPAD. Consequently, the paper entails the same limitations as the data source itself. In fact, ESPAD is a survey conducted only among high school students aged 16; the findings of this study may therefore be not extendable to young people not in education, who tend to report greater adoption of risk behaviors [36,37,38]. Furthermore, future studies might extend the analysis, to include students of different ages. 

Second, it should be mentioned that, in order to perform the current analysis, we implemented a dichotomization when dealing with sublevels within each independent variable. This clearly implies a loss of richness of the information provided by survey respondents. To tackle this, instead of looking only at a generic use, we differentiated between different frequencies of use that we deemed able to identify different patterns of use, i.e., experimental, episodic and frequent use. This allows us to somehow tackle the mentioned loss of richness, as by doing so, users are no longer considered a uniform category, as done in several previous works, and to explore differential associations between their patterns of use and the socioeconomic conditions. 

In addition, in this paper, only cannabis, cocaine and heroin are studied. Future studies should include other addictive substances that may be associated, such as alcohol. Finally, ESPAD entails the common limitations of self-reported data (e.g., issues related to memory recall and social desirability biases, leading to under- or over-reporting of risk behaviors). Although issues of truthfulness are more likely to arise when surveys are administered by personal interview, and in our case, the ESPAD survey is anonymous and self-administered, these concerns have to be mentioned.

## 5. Conclusions

This paper shows that many patterns of use among adolescents are associated with a lower socioeconomic status and lower parental education. In particular, when focusing on frequent use, whilst low SES plays a role in cannabis consumption, low parental education seems to influence the probability of cocaine and heroin use. The association of poor school connectedness and, to a lesser extent, of low individual sociocultural resources with vulnerability for drug use among adolescents was confirmed by our findings, possibly contributing to partially explain the link between socioeconomic characteristics and drug use.

From a policy perspective, the evidence provided in this paper may prove particularly important for the more general issue of inequality of opportunities, suggesting that drug policies should be combined with actions aimed at removing of all barriers to social inclusion that are imputable to the socioeconomic background of adolescents. A priority for future research is to identify effective policy levers that may act in this direction.

## Figures and Tables

**Figure 1 ijerph-17-01306-f001:**
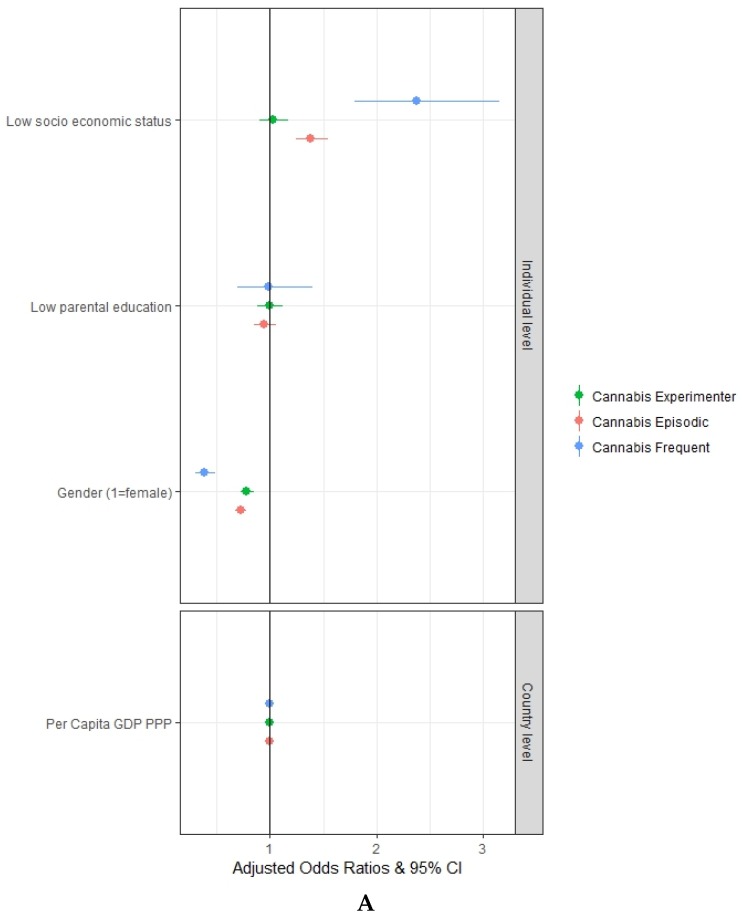
Multivariate multilevel mixed-effects logistic regression adjusted odds ratios with a 95% confidence interval: (**A**) relation between students’ socioeconomic status, parental education and the patterns of cannabis use (Experimenter, Episodic and Frequent); (**B**) relation between students’ socioeconomic status, parental education, truancy at school, reading books for enjoyment and the patterns of cannabis use (Experimenter, Episodic and Frequent). Authors’ elaboration on ESPAD data.

**Figure 2 ijerph-17-01306-f002:**
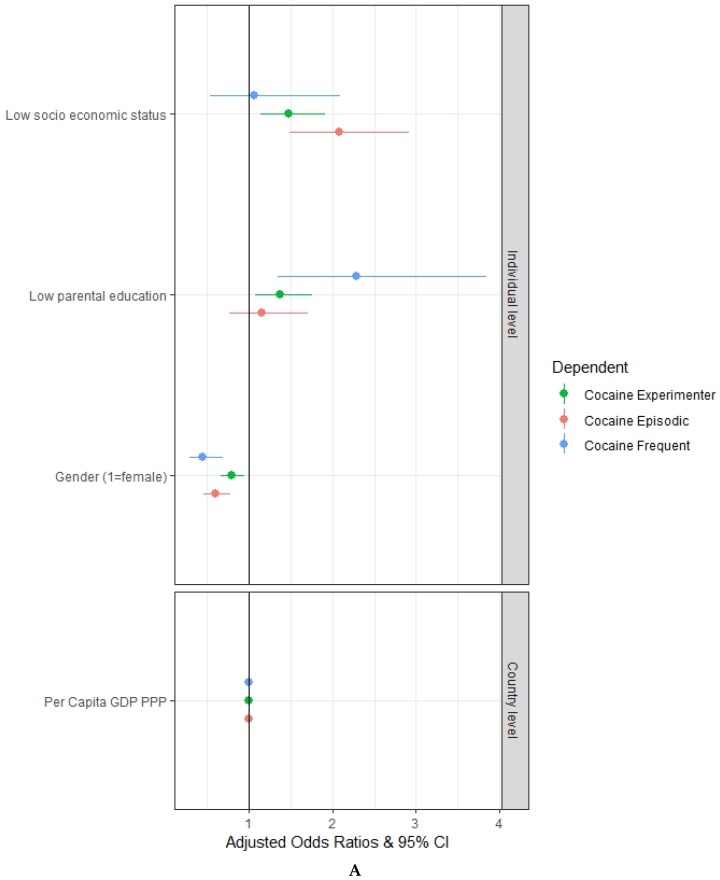
Multivariate multilevel mixed-effects logistic regression adjusted odds ratios with a 95% confidence interval: (**A**) relation between students’ socioeconomic status, parental education and the patterns of cocaine use (Experimenter, Episodic and Frequent); (**B**) relation between students’ socioeconomic status, parental education, truancy at school, reading books for enjoyment and the patterns of cocaine use (Experimenter, Episodic and Frequent). Authors’ elaboration on ESPAD data.

**Figure 3 ijerph-17-01306-f003:**
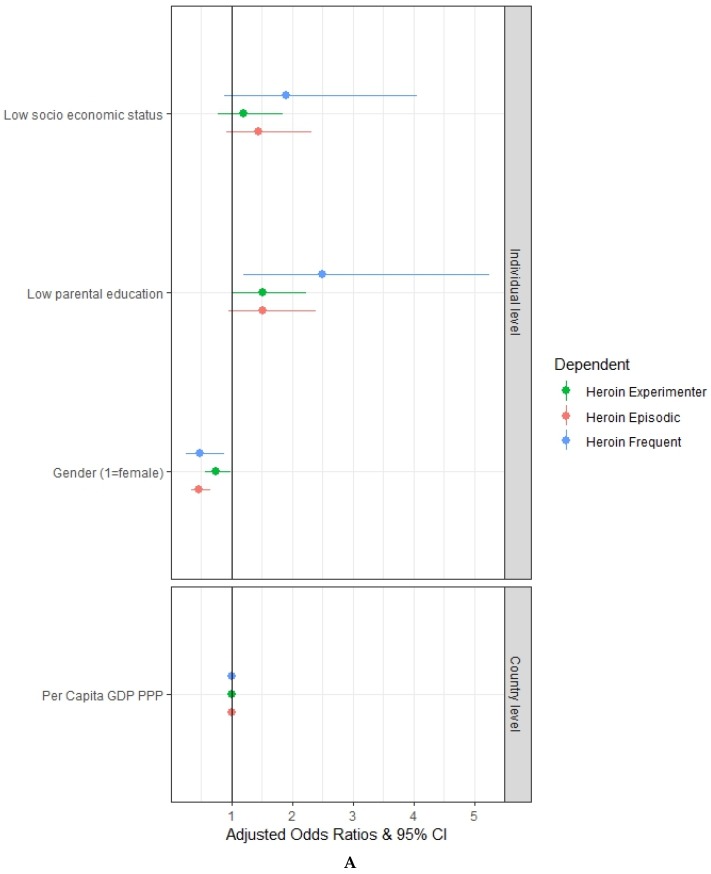
Multivariate multilevel mixed-effects logistic regression adjusted odds ratios with a 95% confidence interval: (**A**) relation between students’ socioeconomic status, parental education and the patterns of heroin use (Experimenter, Episodic and Frequent); (**B**) relation between students’ socioeconomic status, parental education, truancy at school, reading books for enjoyment and the patterns of heroin use (Experimenter, Episodic and Frequent). Authors’ elaboration on ESPAD data.

**Figure 4 ijerph-17-01306-f004:**
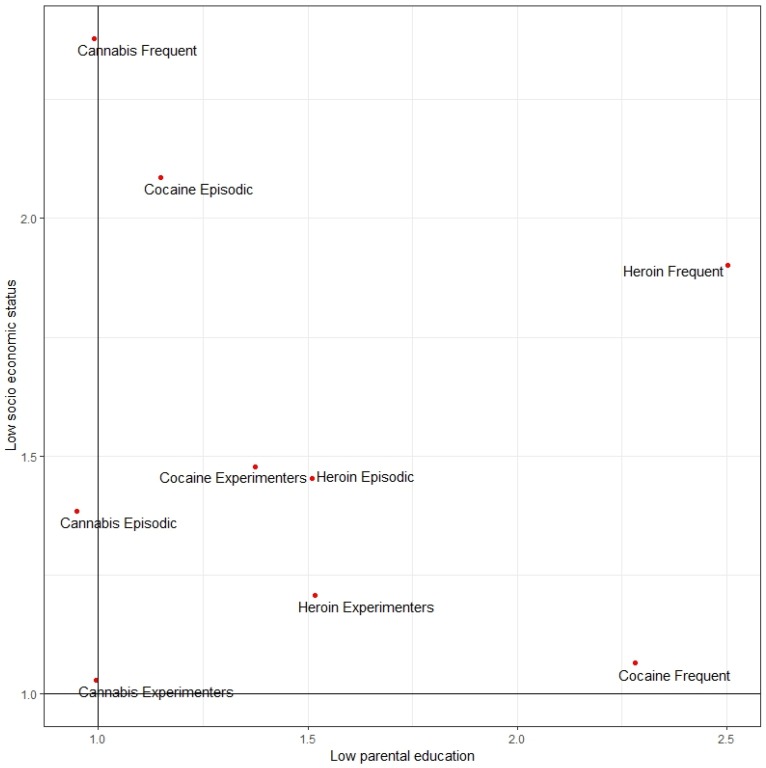
Quadrant analysis with adjusted Odds Ratios (aOR) for low socioeconomic status and low parental education with model A. Authors’ elaboration on ESPAD data. Note: upper right quadrant = low socioeconomic status and low parental education; bottom right quadrant = high socioeconomic status and low parental education; bottom left quadrant = high socioeconomic status and high parental education; upper left quadrant = low socioeconomic status and high parental education.

**Figure 5 ijerph-17-01306-f005:**
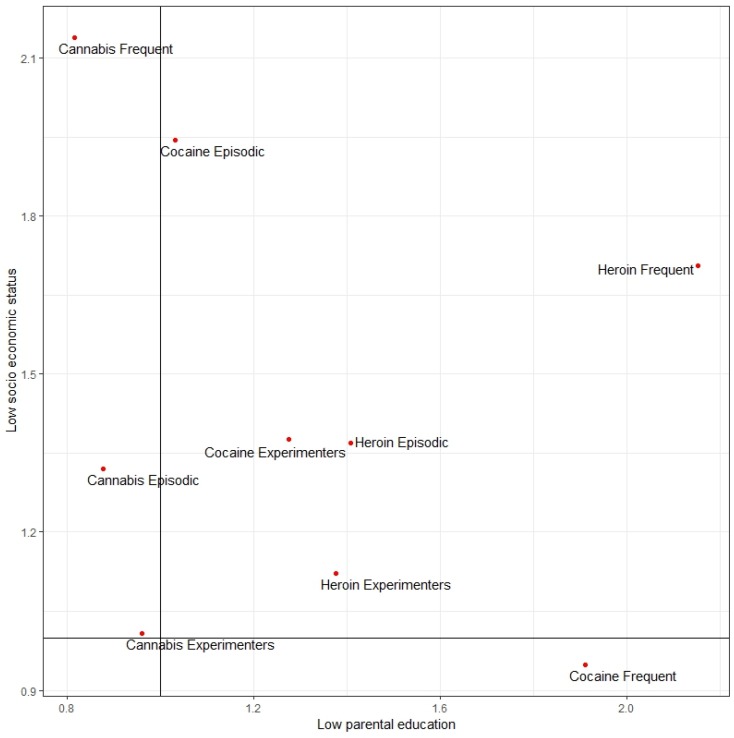
Quadrant analysis with adjusted Odds Ratios (aOR) for low socioeconomic status and low parental education with model B. Authors’ elaboration on ESPAD data. Note: upper right quadrant = low socioeconomic status and low parental education; bottom right quadrant = high socioeconomic status and low parental education; bottom left quadrant = high socioeconomic status and high parental education; upper left quadrant = low socioeconomic status and high parental education.

**Table 1 ijerph-17-01306-t001:** Descriptive statistics.

Country	Sample Size			Cannabis Use									Cocaine Use									Heroin Use								
				Episodic			Experimenter			Frequent			Episodic			Experimenter			Frequent			Episodic			Experimenter			Frequent		
	T	M	F	T	M	F	T	M	F	T	M	F	T	M	F	T	M	F	T	M	F	T	M	F	T	M	F	T	M	F
Albania	1758	800	958	2.67	5.00	0.73	2.84	5.25	0.84	0.46	0.75	0.21	0.51	0.88	0.21	1.42	2.38	0.63	0.40	0.88	0.00	0.46	1.00	0.00	0.51	0.63	0.42	0.11	0.00	0.21
Austria	1612	756	856	9.88	11.52	8.17	8.69	9.85	7.48	1.41	1.69	1.12	0.47	0.52	0.42	1.46	1.09	1.83	0.15	0.00	0.31	0.08	0.15	0.00	0.20	0.00	0.42	0.00	0.00	0.00
Belgium	1145	582	563	10.35	13.75	6.93	6.27	7.39	5.15	0.52	0.86	0.18	0.61	0.69	0.53	1.92	1.72	2.13	0.09	0.00	0.18	0.00	0.00	0.00	0.61	0.69	0.53	0.00	0.00	0.00
Bulgaria	2066	970	1096	15.00	14.54	15.42	10.12	11.34	9.03	1.55	2.37	0.82	0.92	1.34	0.55	2.42	3.20	1.73	0.53	0.72	0.36	0.73	1.13	0.36	1.50	1.96	1.09	0.24	0.52	0.00
Croatia	1783	910	873	11.10	11.54	10.65	9.20	10.55	7.79	0.90	1.21	0.57	0.17	0.22	0.11	1.01	0.55	1.49	0.22	0.33	0.11	0.39	0.55	0.23	0.45	0.22	0.69	0.06	0.00	0.11
Cyprus	1438	655	783	2.02	2.90	1.28	2.43	3.66	1.40	0.90	1.07	0.77	0.76	1.37	0.26	0.83	1.22	0.51	0.35	0.46	0.26	0.83	1.68	0.13	0.63	0.92	0.38	0.21	0.15	0.26
Czech Rep.	2221	1035	1186	19.48	18.73	20.20	14.73	13.76	15.67	1.29	1.87	0.74	0.12	0.20	0.04	0.92	0.91	0.93	0.07	0.15	0.00	0.19	0.20	0.18	0.37	0.62	0.13	0.00	0.00	0.00
Faroe Isl.	223	112	111	1.35	1.79	0.90	4.04	3.57	4.50	0.00	0.00	0.00	0.00	0.00	0.00	0.00	0.00	0.00	0.00	0.00	0.00	0.00	0.00	0.00	0.00	0.00	0.00	0.45	0.00	0.90
France	1605	762	843	19.97	22.59	17.52	9.63	9.88	9.40	1.62	2.03	1.24	1.28	1.08	1.47	2.14	2.79	1.53	0.40	0.25	0.53	0.31	0.23	0.38	0.83	0.81	0.84	0.53	0.43	0.63
Georgia	1239	655	839	6.46	2.60	1.55	4.60	4.27	0.60	0.48	0.31	0.12	0.24	0.31	0.12	1.21	0.31	0.24	0.40	0.46	0.00	1.45	0.00	0.24	0.24	0.15	0.12	0.16	0.31	0.00
Germany	628	634	605	12.14	12.15	0.50	8.33	7.10	1.98	0.81	0.79	0.17	0.40	0.47	0.00	1.16	1.42	0.99	0.28	0.47	0.33	0.00	2.21	0.66	0.83	0.47	0.00	0.00	0.16	0.17
Greece	2370	294	334	3.19	16.63	8.22	4.18	9.75	7.09	0.42	1.32	0.38	0.30	0.43	0.38	0.48	0.99	1.31	0.20	0.61	0.00	0.24	0.00	0.00	0.14	1.34	0.38	0.02	0.00	0.00
Hungary	1987	1154	1216	5.38	4.21	2.21	6.49	5.46	2.94	0.05	0.86	0.00	0.50	0.58	0.03	1.48	0.59	0.37	0.00	0.40	0.00	0.24	0.35	0.13	0.41	0.09	0.18	0.00	0.03	0.00
Iceland	1801	987	1000	3.55	6.48	4.28	1.89	6.32	6.66	0.28	0.10	0.00	0.56	0.61	0.40	0.78	1.46	1.51	0.17	0.00	0.00	0.28	0.10	0.37	0.11	0.51	0.32	0.00	0.00	0.00
Ireland	873	864	937	9.39	4.17	2.99	6.19	2.20	1.60	1.37	0.23	0.32	0.69	0.46	0.64	1.03	0.69	0.85	0.23	0.23	0.11	0.46	0.23	0.32	0.34	0.12	0.11	0.00	0.00	0.00
Italy	2805	428	445	16.65	11.92	6.97	7.81	6.54	5.84	1.82	2.10	0.67	0.50	1.17	0.22	1.53	1.40	0.67	0.32	0.23	0.22	0.39	0.93	0.00	0.78	0.70	0.00	0.25	0.00	0.00
Latvia	560	1373	1432	7.46	19.16	14.25	5.32	8.16	7.47	0.35	2.99	0.70	0.25	0.66	0.35	0.52	1.75	1.33	0.57	0.66	0.00	0.36	0.36	0.42	0.23	0.87	0.70	0.15	0.44	0.07
Liechtenstein	98	250	310	21.43	8.90	6.20	12.24	5.64	5.03	1.02	0.75	0.00	1.02	0.34	0.18	0.00	1.11	0.00	0.00	1.22	0.00	0.00	0.49	0.25	0.00	0.50	0.00	0.00	0.31	0.00
Lithuania	1392	50	48	7.04	28.00	14.58	9.27	18.00	6.25	0.22	2.00	0.00	0.22	2.00	0.00	1.44	0.00	0.00	0.14	0.00	0.00	0.07	0.00	0.00	0.65	0.00	0.00	0.07	0.00	0.00
Macedonia	1494	686	706	2.01	8.02	6.09	2.21	11.22	7.37	0.20	0.44	0.00	0.20	0.15	0.28	0.27	1.46	1.42	0.20	0.15	0.14	0.13	0.15	0.00	0.13	0.73	0.57	0.13	0.15	0.00
Malta	2024	964	1060	5.78	5.60	5.94	5.09	5.39	4.81	0.64	0.62	0.66	0.69	0.41	0.94	1.38	1.24	1.51	0.05	0.10	0.00	0.10	0.10	0.09	0.15	0.21	0.09	0.10	0.21	0.00
Moldova	1556	774	782	1.35	2.20	0.51	2.83	4.39	1.28	0.13	0.26	0.00	0.13	0.13	0.13	0.32	0.52	0.13	0.00	0.00	0.00	0.06	0.13	0.00	0.19	0.39	0.00	0.00	0.00	0.00
Montenegro	3008	1478	1530	3.46	5.21	1.76	3.19	3.99	2.42	0.33	0.47	0.20	1.20	1.56	0.85	1.23	2.03	0.46	0.30	0.34	0.26	0.80	1.15	0.46	0.60	0.88	0.33	0.20	0.27	0.13
Netherlands	1125	570	555	14.52	17.27	11.83	7.61	6.49	8.69	1.15	1.80	0.52	0.64	1.13	0.17	0.66	0.85	0.48	0.19	0.00	0.38	0.29	0.17	0.41	0.18	0.37	0.00	0.08	0.17	0.00
Poland	7695	3636	4059	11.80	14.57	9.31	9.60	10.79	8.53	0.67	0.86	0.50	0.97	1.08	0.88	2.08	2.29	1.89	0.18	0.25	0.11	0.92	1.15	0.71	1.25	1.30	1.21	0.21	0.19	0.24
Romania	2098	977	1121	2.72	3.28	2.23	3.67	4.71	2.77	0.19	0.31	0.09	0.76	0.72	0.80	1.95	1.74	2.14	0.33	0.41	0.27	0.19	0.20	0.18	0.86	1.02	0.71	0.19	0.41	0.00
Slovenia	2492	1165	1327	12.60	12.62	12.58	9.75	10.13	9.42	1.32	1.97	0.75	0.44	0.60	0.30	1.24	0.69	1.73	0.16	0.26	0.08	0.20	0.17	0.23	0.48	0.17	0.75	0.00	0.00	0.00
Ukraine	1340	615	725	3.68	4.55	2.96	4.99	7.12	3.26	0.12	0.27	0.00	0.18	0.40	0.00	0.82	0.97	0.70	0.12	0.15	0.10	0.25	0.43	0.10	0.26	0.13	0.36	0.00	0.00	0.00
Average	1801	862	939	8.66	10.35	7.03	6.54	7.60	5.55	0.72	1.08	0.38	0.53	0.70	0.37	1.13	1.26	1.02	0.22	0.31	0.13	0.34	0.47	0.21	0.46	0.56	0.37	0.11	0.13	0.10

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
