# Peer review of "Socioeconomic Status, Parental Education, School Connectedness and Individual Socio-Cultural Resources in Vulnerability for Drug Use among Students"

_ijerph, 2020, doi:10.3390/ijerph17041306_

Round 1
Reviewer 1 Report
The authors conducted an interesting analysis of ESPAD 2015 data, from which they derive useful indications for knowing the factors associated with addictions and guiding prevention measures.
The paper lacks the Limitations section, as well as prospects for future research.
Among the limitations it must be discussed that studies are based on a single data source. Consequently, the paperentails the same limitations as the data source itself.
The authors explored only cocaine and heroin, without studying other addictions that may be associated, such as alcohol.
The work indicates many points that need clarification and that could be addressed in the future.
Author Response
Point 1: The paper lacks the Limitations section, as well as prospects for future research.
Response 1: Thank you, we added a discussion of the limitations (Discussion section, as indicated in the guidelines for authors) and of the prospects for future research (Conclusions section).
Point 2: Among the limitations it must be discussed that studies are based on a single data source. Consequently, the paper entails the same limitations as the data source itself.
The authors explored only cocaine and heroin, without studying other addictions that may be associated, such as alcohol.
The work indicates many points that need clarification and that could be addressed in the future.
Response 2: Thank you, we added also these issues to the discussion of the limitations.

Reviewer 2 Report
Re: Peer-Review Report
Dear Authors,
I have found the manuscript to be novel in terms of study implications, and statistical analytics, including the utilization of two collateral models of multivariate multi-level mixed-effects logistic regression and a graphical presentation of quadrant analysis in connection with substance use. I have recommended the authors to do minor revisions.
I have found the manuscript to be suitable for publication, pending minor revisions. All the revisions fit the category of "corrections to minor methodological errors and text editing". The authors should comply with each of the following:
-The introduction section is well-written. However, the authors should delete Line-65 and 66 to avoid redundancy.
-The Materials and Methods are well-presented and partitioned based on the independent and dependent variables included in the study design and statistical analysis. However, the authors should provide a statement in connection with ethics and ethical approval and the relevant IRB and supporting documentation to validate this aspect for the journal’s editor.
-The authors implemented dichotomization as an over-simplification approach when dealing with sub-levels within each independent variable.
-Line-115 and 121, please, provide a reference from the existing literature to justify the rationale for including these two additional independent variables in model-2 of your statistical analysis. It is mandatory, as all subsequent work within the study relying on it.
-Page-4, please, delete the footnote concerning R, as it is well-known worldwide. Instead, you can include it within the reference list.
-Overall, the results section is well-written and organized based on the dependent variable, i.e., type of substance use. The 4th section of the results, a “global perspective”, is a bonus addition with proper implementation of visual illustration for the Quadrant analysis, congratulations.
-Concerning acronyms, please explain each, for example, the “GDP” (at Line-283).
-The pixelated quality of the first three figures is poor. Provide high-resolution and editable versions of those figures (Fig. 1 to 3).
-The first three figures also included adjusted OR for “Per Capita GDP” and “Gender”. The authors included neither of these two in the discussion. Besides, the rationale for implementing each in the statistical analysis should be clearly explained, particularly for gender, and the authors did not pay much attention to it in all three sections (Materials & Methods, Results, and Discussion) while being included in the Figures.
-Figure-4 is deploying the first model, i.e., without the inclusion of “correctness at school” and reading books”, while adjusting for gender and per-capita GDP. The authors should provide a collateral comparison based on model-2, implementing all four independent variables.
-The discussion section is the weakest in your article, and you should further expand and enrich with 1) Additional discussion of your results. 2) Discussion and contrasting with the literature.
-The conclusion section is right. However, I find it unusual to find a reference within it (Line-325, reference no. 27). Please, edit and incorporate it into the discussion section.
-The references are excellent, from reliable resources, and up-to-date. However, please re-write with precision following the referencing style adopted by the International Journal of Environmental Research and Public Health [available at https://www.mdpi.com/journal/ijerph/instructions#references].
Thank you for your efforts.

Author Response
Point 1: The introduction section is well-written. However, the authors should delete Line-65 and 66 to avoid redundancy.
Response 1: Thank you, Line-65 and 66 have been deleted.
Point 2: The Materials and Methods are well-presented and partitioned based on the independent and dependent variables included in the study design and statistical analysis. However, the authors should provide a statement in connection with ethics and ethical approval and the relevant IRB and supporting documentation to validate this aspect for the journal’s editor.
Response 2: Thank you, a statement in connection with ethics has been provided in Line 81-83 and specific reference has been added for more details about ethics in each country participating in ESPAD in 2015.
Point 3: The authors implemented dichotomization as an over-simplification approach when dealing with sub-levels within each independent variable.
Response 3: Thank you for your comment. In order to perform the current analysis, we implemented a dichotomization when dealing with sub-levels within each independent variable. It is true that implementing a dichotomization clearly implies a loss of richness of the information provided by survey respondents. To tackle this, instead of looking only at a generic use (e.g. using the lifetime use variable and dichotomizing into 1 for any use in the life and 0 otherwise), we differentiated between different frequencies of use that we deem can identify different “patterns of use”. The frequencies of use of the investigated substances were then dichotomized to create an index of use (yes/no) for each pattern, i.e. experimental, episodic and frequent use. This allows to somehow tackle the mentioned loss of richness, as by doing so users are no longer considered an uniform category, as done in several previous works, and to explore differential associations between their patterns of use and the socio-economic conditions.
We added this limitation at the end of the Discussion section.
Point 4: Line-115 and 121, please, provide a reference from the existing literature to justify the rationale for including these two additional independent variables in model-2 of your statistical analysis. It is mandatory, as all subsequent work within the study relying on it.
Response 4: Following your indications, we have added the following sentence in the (old) line 115:
“School connectedness should be taken into account since previous studies have shown that the adoption of risk behaviours among youth is significantly linked to school absenteeism [17, 19].”
We have also added the following sentence in the old line 121:
“It has been shown that socio-economic status heavily influences access to relevant networks (e.g. internet, newspapers, libraries) for socio-cultural resources [23] which in turn seems to become an additional risk factor for substance use [9, 12].”
Point 5: Page-4, please, delete the footnote concerning R, as it is well-known worldwide. Instead, you can include it within the reference list.
Response 5: Thank you, it has been done.
Point 6: Concerning acronyms, please explain each, for example, the “GDP” (at Line-283).
Response 6: Thank you, for the acronyms “PISA” and “OECD” at Lines 107-109 and “GDP” at Line 293 the full name has been provided.
Point 7: The pixelated quality of the first three figures is poor. Provide high-resolution and editable versions of those figures (Fig. 1 to 3).
Response 7: Thank you, we did this.
Point 8: The first three figures also included adjusted OR for “Per Capita GDP” and “Gender”. The authors included neither of these two in the discussion. Besides, the rationale for implementing each in the statistical analysis should be clearly explained, particularly for gender, and the authors did not pay much attention to it in all three sections (Materials & Methods, Results, and Discussion) while being included in the Figures.
Response 8: Thank you, the rationale for implementing “Per Capita GDP” and “Gender” in the statistical analysis has been clearly explained in Lines 222-227 of the Materials & Methods and a discussion included in Lines 362-363 of the Discussion section.
Point 9: Figure-4 is deploying the first model, i.e., without the inclusion of “correctness at school” and reading books”, while adjusting for gender and per-capita GDP. The authors should provide a collateral comparison based on model-2, implementing all four independent variables.
Response 9: Done, we have npw added Figure 5 and te following sentences in the results section:
“In Figure 5 the same analysis shown in Figure 4 is repeated using the aORs described in Figures 1.B, 2.B and 3.B (model with the inclusion of “truancy at school” and “reading books”, in addition to gender and per-capita GDP among independent variables). Overall, the same evidences shown in Figure 4 are confirmed in Figure 5, but one exception is worthy of attention: the frequent use of cocaine is associates with high socio economic status with model B. Although not significant (see Figure 2B), this may be attributed to the high price of cocaine, making it not affordable for youth from family with low socio-economic status [30]. “
Point 10: The discussion section is the weakest in your article, and you should further expand and enrich with 1) Additional discussion of your results. 2) Discussion and contrasting with the literature.
Response 10: Thank you, we added an additional discussion of results, linking it to a discussion and contrasting with the literature, in particular related to previous findings drawn from the ESPAD data. We also included a discussion of the limitations of the study.
Point 11: The conclusion section is right. However, I find it unusual to find a reference within it (Line-325, reference no. 27). Please, edit and incorporate it into the discussion section.
Response 11: Thank you, the reference has been removed from the Conclusions section and incorporated it into the Discussion section, Line-359.
Point 12: The references are excellent, from reliable resources, and up-to-date. However, please re-write with precision following the referencing style adopted by the International Journal of Environmental Research and Public Health [available at https://www.mdpi.com/journal/ijerph/instructions#references].
Response 12: Done.

Reviewer 3 Report
The manuscript submitted by Gerra and coworkers evaluated the role ofseveral social factors in the vulnerability for drug use among
students by analysis the data obtained Results from the European
School Survey Project on Alcohol and Other Drugs (ESPAD) in 2015.
The firs important issue is the novelty of these results. Many papers have been published on this interaction
"social factors and drug abus in stduents" and several reports (see belo)
have been published form ESPAD study.
1: Perelman J, Leão T, Kunst AE. Smoking and school absenteeism among 15- to 16-year-old adolescents: a cross-section analysis on 36 European countries. Eur J Public Health. 2019 Aug 1;29(4):778-784. doi: 10.1093/eurpub/ckz110. PubMed PMID: 31168621; PubMed Central PMCID: PMC6660109. 2: Shackleton N, Milne BJ, Jerrim J. Socioeconomic Inequalities in Adolescent Substance Use: Evidence From Twenty-Four European Countries. Subst Use Misuse. 2019;54(6):1044-1049. doi: 10.1080/10826084.2018.1549080. Epub 2019 Jan 16. PubMed PMID: 30648460. 3: König C, Skriver MV, Iburg KM, Rowlands G. Understanding Educational and Psychosocial Factors Associated with Alcohol Use among Adolescents in Denmark; Implications for Health Literacy Interventions. Int J Environ Res Public Health. 2018 Aug 6;15(8). pii: E1671. doi: 10.3390/ijerph15081671. PubMed PMID: 30082674; PubMed Central PMCID: PMC6121249. 4: Raitasalo K, Simonen J, Tigerstedt C, Mäkelä P, Tapanainen H. What is going on in underage drinking? Reflections on Finnish European school survey project on alcohol and other drugs data 1999-2015. Drug Alcohol Rev. 2018 Apr;37 Suppl 1:S76-S84. doi: 10.1111/dar.12697. Epub 2018 Mar 23. PubMed PMID: 29573
In addition a book about ESPAD results has been published and is freely available
in internet:
https://www.researchgate.net/publication/308341566_ESPAD_Report_2015_Results_from_the_European_School_Survey_Project_on_Alcohol_and_Other_Drugs
A careful analysis of the novelty of these finding should be provided from teh Introduction to
the Discussion section.
A second crucial issue requires a methodological approach: is the sample of stduents representative
of each country? please provide such information and calculation of sample size
for each country.
Once evaluate the previous issue authors would need to correct the analysis of data
taking into account the fact that some population is overrepresented in
teh ESPAD study while for other countries did not.
The adjusted ODDS ratios shoudl taking into account the previous results and
the global ORs should also be analysed taking into account sample size and
representativness for the country population.
A statystical analysis among countries should be provided for each drug of abuse.
Figures have very poor resolution.
Author Response
Point 1: The firs important issue is the novelty of these results. Many papers have been published on this interaction “social factors and drug abus in stduents" and several reports (see belo) have been published form ESPAD study. 1: Perelman J, Leão T, Kunst AE. Smoking and school absenteeism among 15- to 16-year-old adolescents: a cross-section analysis on 36 European countries. Eur J Public Health. 2019 Aug 1;29(4):778-784. doi: 10.1093/eurpub/ckz110. PubMed PMID: 31168621; PubMed Central PMCID: PMC6660109. 2: Shackleton N, Milne BJ, Jerrim J. Socioeconomic Inequalities in Adolescent Substance Use: Evidence From Twenty-Four European Countries. Subst Use Misuse. 2019;54(6):1044-1049. doi: 10.1080/10826084.2018.1549080. Epub 2019 Jan 16. PubMed PMID: 30648460. 3: König C, Skriver MV, Iburg KM, Rowlands G. Understanding Educational and Psychosocial Factors Associated with Alcohol Use among Adolescents in Denmark; Implications for Health Literacy Interventions. Int J Environ Res Public Health. 2018 Aug 6;15(8). pii: E1671. doi: 10.3390/ijerph15081671. PubMed PMID: 30082674; PubMed Central PMCID: PMC6121249. 4: Raitasalo K, Simonen J, Tigerstedt C, Mäkelä P, Tapanainen H. What is going on in underage drinking? Reflections on Finnish European school survey project on alcohol and other drugs data 1999-2015. Drug Alcohol Rev. 2018 Apr;37 Suppl 1:S76-S84. doi: 10.1111/dar.12697. Epub 2018 Mar 23. PubMed PMID: 29573. In addition a book about ESPAD results has been published and is freely available in internet: https://www.researchgate.net/publication/308341566_ESPAD_Report_2015_Results_from_the_European_School_Survey_Project_on_Alcohol_and_Other_Drugs. A careful analysis of the novelty of these finding should be provided from the Introduction to the Discussion section.
Response 1: Thank you, we have now included the following sentences at the end of the Introduction:
“Considering papers which analyze the interaction between social factors and psychoactive substance abuse in ESPAD, Perelman et al. [17] have shown an association between heavy smoking and school absenteeism among youth. Shackleton et al. [18] have proved that, although there are large country level differences in socioeconomic inequalities and adolescent substance use, lower socioeconomic status had significantly higher odds of heavy episodic drinking, regular smoking, and cannabis use. Considering only the Danish students, König et al. [19] showed that higher school performance was related to lower alcohol consumption, but low socio-demographic status was not associated with higher alcohol consumption. Using the Finnish surveys from 1999 to 2015, Raitasalo et al. [20] evidence that the decline in alcohol use and heavy episodic drinking among youth is associated with the fact that obtaining alcohol has become more difficult, there has been an increasing in the parental monitoring, and there has been the introduction of new digital technologies and new forms of interaction within families and peer groups. Compared with these previous evidences on ESPAD, the distinction of three different patterns of use (experimental, episodic and frequent) of three different illicit drugs (cannabis, cocaine, and heroin) in the present study allows us to analyze the differentiated association that social factors may have with the students’ differentiated behaviors. As the students’ experimental, episodic, and frequent use of cannabis, cocaine, and heroin are heterogeneous risk behaviors, this study allows to capture a differentiated association with socioeconomic status of the family, parental education, school connectedness, and individual socio-cultural resources.”
And we have now included references to these studies along with the text in the Discussion section.
Point 2: A second crucial issue requires a methodological approach: is the sample of stduents representative of each country? please provide such information and calculation of sample size
for each country. Once evaluate the previous issue authors would need to correct the analysis of data taking into account the fact that some population is overrepresented in the ESPAD study while for other countries did not. The adjusted ODDS ratios should taking into account the previous results and the global ORs should also be analysed taking into account sample size and representativeness for the country population. A statistical analysis among countries should be provided for each drug of abuse.
Response 2: As far as the representativeness is concerned, we hae added the following sentence into the Methos section soon after the ESPAD data is mentioned:
“Participating countries adhered to common research guidelines to guarantee consistency in sampling, questionnaires, and survey implementation, and conformed to the respective national ethics and data protection regulation.”
Sample size for each country is now provided in the second column in Tale 1. The analysis is already corrected for country-level effect since we used a multivariate multi-level mixed-effects logistic regression. To make it more clear we added the following sentence into the Method section:
“Models were performed on the overall sample, modelling different countries as random effects as in Molinaro et al. [24].”
A statistical analysis among countries is provided for each drug of abuse in Table 1.
Point 3: Figures have very poor resolution.
Response 3: We have now provided all figures in separate files both in .jpeg and .svg format

Round 2
Reviewer 3 Report
The authors did not provide the calculation of sample size for each country thus teh generalization of the present results may not be valid for each country. Authors reported in the Table 1 only the sample size. It's important to know if the sample of students in each country is representative or not for the students' population of that country.
Gender differences should be represented.
The Figures have still a poor resolution.
Author Response
Point 1: The authors did not provide the calculation of sample size for each country thus teh generalization of the present results may not be valid for each country. Authors reported in the Table 1 only the sample size. It's important to know if the sample of students in each country is representative or not for the students' population of that country.
Response 1: We would like to draw the attention of the reviewer on the fact the the ESPAD project is a cross-national survey performed on national samples of 16 years-old students in each participating country. The target population of the ESPAD study is defined as the national population of students who turn 16 during the calendar year of the survey, excluding those who were enrolled in either special schools or special classes for students with learning disorders or severe physical disabilities.
By definition, in order to participate in ESPAD these samples have to be representative at country level.
We cannot therefore provide the exact calculation of the numerosity necessary in each country, as this is something that is evaluated by each national Principal Investigator based on her/his methodology. It is important to keep in mind that the results for Cyprus, Moldova and Belgium are representative only for the populations from which the samples were drawn, according to the geographical limitations mentioned in the report.
In order to clarify the point raised by the reviewer, we included the following in Lines 101-102:
“Participating countries adhered to common research guidelines to guarantee consistency in sampling, questionnaires, and survey implementation, and confirmed to the respective national ethics and data protection regulation. All samples are nationally representative, apart from Belgium (only the Flanders region), Cyprus (only government-controlled areas) and Moldova (the Transnistria region is not included). Details about sampling and, data collection methodology and ethics in each country are reported in Kraus et al. [21] and Guttormsson et al. [22]. An overview of the geographical coverage, sampling procedure in each country, representativeness of the samples and caractheristics of the samples is provided in Tables C and F - H, of the ESPAD methodology Report [22, pp. 10, 16, 18, 29-30].”
We hope that this specification, paired with the reference to the two 2015 ESPAD Reports (Results and Methodological ones), were all the above detailed information and statistics are provided, is able to provide enough clarity to the reader as to the representativeness of the samples and the generalizability of the results of the paper.
Point 2: Gender differences should be represented.
Response 2: We have now included Gender differences in new Table 1
Point 3: The Figures have still a poor resolution.
Response 3: We produced high resolution Figures both in .jpeg and .svg format, we sent them to the editor by email. We have now also included them (the .jpeg) into the text.

Round 3
Reviewer 3 Report
8